# Thoracic Tumor Associated with a Unilateral Empyema in a Beef Cow: A Case Report

**DOI:** 10.3390/vetsci10060376

**Published:** 2023-05-27

**Authors:** Rodolphe Robcis, Charly De Campos, Bénédicte Garapin, Marie-Noëlle Lucas, Agnès Poujade, Nicolas Gaide, Maxence Delverdier, Renaud Maillard

**Affiliations:** 1Clinic of Ruminants, Ecole Nationale Vétérinaire de Toulouse, 31300 Toulouse, France; charly.decampos@envt.fr (C.D.C.); renaud.maillard@envt.fr (R.M.); 2Department of Basic Sciences, IHAP, INRAE, ENVT, Université de Toulouse, 31013 Toulouse, France; benedicte.garapin@envt.fr (B.G.); nicolas.gaide@envt.fr (N.G.); maxence.delverdier@envt.fr (M.D.); 3Laboratoire d’Anatomie Pathologique Vétérinaire du Sud-Ouest, 31100 Toulouse, France; mn.lucas@lapvso.fr; 4Necropsy Platform, Ecole Nationale Vétérinaire de Toulouse, 31300 Toulouse, France; agnes.poujade@envt.fr

**Keywords:** tumor, sarcoma, cow, cattle, case report

## Abstract

**Simple Summary:**

Tumors are a very rare condition in cattle. Moreover, they are poorly described in the literature, especially in bovine species. The aim of this study was to accurately describe a procedure leading to the diagnosis of thoracic sarcoma in a beef cow. The procedure started from the initial clinical signs motivating specific investigations (blood biochemistry, cytology, thoracic ultrasonography, and ultrasound-guided biopsy) and resulted in interesting laboratory findings with histology and targeted immunochemistry. The concomitant support treatment is also thoroughly provided in the manuscript. This study may offer valuable support for veterinary practitioners in future similar cases.

**Abstract:**

Tumors in cows are not frequently reported in the literature. They often represent unusual findings in live animals and are incidental at slaughter with rare positive therapeutic outcomes for farmers. A 9-year-old beef cow was referred to the hospital of ruminants of the National Veterinary School of Toulouse, France. The cow started to become sick 10 days prior, and major symptoms were anorexia, arched back, tachycardia, and tachypnea associated with significantly attenuated cardiac and pulmonary sounds upon right-sided auscultation. After specific investigations, a thoracic sarcoma associated with unilateral empyema was diagnosed. The empyema was treated, and supportive treatment was only performed for the tumor. Although the sarcoma remained, clinical improvement was significant, and the cow went back to her farm of origin. After the end of the withdrawal period, the cow recovered clinically but was culled by the owners for economic reasons. The present case report offers a continuum from the initial clinical signs motivating specific investigations to interesting laboratory findings, which were confirmed post-mortem.

## 1. Introduction

Pleural disorders are common in bovine medicine. Among them, inflammatory conditions, known as pleuritis, are the most frequent. The etiology is mainly bacterial, including the extension of exudative bronchopneumonia, traumatic reticuloperitonitis, septicemia, and zoonotic tuberculosis [1,2,3].

Tumors are a very rare condition in cattle. The prevalence of tumors is estimated at a very low rate of 0.06% in adults and 0.006% in calves [4]. The incidence varies with localization and the affected organs and tissues. Thoracic tumors are rare, and they represent an unusual finding in live animals and slaughterhouses [5].

Primary malignant neoplasms are classified according to their origin [6]. Carcinomas refer to malignant neoplasms of epithelial cells. Carcinomas are divided into two major subtypes: adenocarcinoma, which develops in an organ or gland, and squamous cell carcinoma, which originates in the squamous epithelium. Sarcomas refer to cancers that originate in mesenchymal cells at the origin of supportive and connective tissues, such as bones, tendons, cartilage, muscle, and fat. They include osteosarcomas, chondrosarcomas, hemangiosarcomas, malignant histiocytosis, lymphomatoid granulomatosis, granular cell tumors, and mesothelioma [6].

Several cases of thoracic sarcomas are well-described in the literature, but to the best of our knowledge, no undifferentiated sarcomas in the thoracic cavity have been reported. Only two undifferentiated sarcomas have been reported. The first was characterized by a multicentric form in the maxillary bone, lung, kidney, spleen, and muscles of the fore- and hindlimbs [7]. The second was a congenital suborbital mass infiltrating the underlying muscle and bone [8].

Here, we describe a clinical case of thoracic sarcoma associated with unilateral empyema in a beef cow.

## 2. Detailed Case Description

### 2.1. Case Presentation and Gross Findings

A 9-year-old beef cow, Limousine breed, was referred to the clinic of ruminants of the National Veterinary School of Toulouse, France. The cow was not pregnant and had calved four months prior. Clinical signs began 10 days before referral, with moderate hyperthermia (39.5–39.6 °C), dry feces, decreased rumination and appetite, an arched back, and attenuated sounds upon heart auscultation. The cow was first treated by the referring field veterinarian. A 5-day antimicrobial treatment was performed (amoxicillin trihydrate, Vetrimoxin 48 heures, CEVA©, Marseille, France, 15 mg/kg of body weight) and two injections of corticosteroids (dexamethasone phosphate, Dexadreson, MSD Animal Health©, Beaucouzé, France, 0.06 mg/kg of body weight) at a 48 h interval. No clinical improvement was noticed by the farmer, justifying the hospitalization at the clinic of ruminants of the National Veterinary School of Toulouse, France.

On arrival, a distant examination of the cow revealed discrete depressed behavior with a correct overall conformation (estimated body condition score of 3.5 according to the 5-scale classification). She weighed 738 kg, which is in accordance with an adult beef cow, even though the owner reported a decrease in weight since the clinical signs occurred. The cow presented complete inappetence. The low rumen fill score, estimated at 2, indicated that the lack of appetite had lasted for several days. An abnormal posture was observed with a persistent arched back, probably due to a non-specific painful phenomenon. Pronounced edema localized at the dewlap was also noticed, which was associated with a bilateral turgidity of the jugular veins. The jugular compression test was applied to jugular veins and was positive in both (i.e., the absence of blood flushing on the segment between the compression point and the heart), indicating a disorder of venous circulation.

Close examination revealed pale ocular, oral, and vaginal mucous membranes accompanied by severe tachycardia (105 beats per minute) and marked tachypnoea (44 movements per minute). The type of breathing was costo-abdominal, and the amplitude was normal. However, right-sided pulmonary auscultation revealed attenuated sounds. Heart examination revealed a duplication of the first cardiac sound (S1). Moreover, left-sided cardiac auscultation was distinctly audible, whereas right-sided auscultation revealed attenuated sounds. Neither rumination movements nor rumination sounds were detected upon observation and auscultation, respectively. A skin fold pinch test in the wither area was positive with no induced curvature, indicating pain located in the thorax and/or cranial part of the abdomen, in accordance with the abnormal posture described above. The rectal temperature (38.7 °C) was in the expected range. The palpation of the external lymph nodes revealed moderate hypertrophy of the right superficial cervical lymph node. Transrectal palpation did not reveal any abnormalities in the urogenital and digestive compartments.

The macroscopic examination of urine collected through spontaneous micturition upon admission did not show any alteration. The defecation process was without abnormality, and the feces were in normal quantity with a correct macroscopic aspect.

The major clinical signs concerned the thoracic organs, especially those on the right side. Differentials included cardiac diseases (exudative pericarditis, endocarditis, or myocardium-centered mass), right-sided thoracic liquid collection (unilateral empyema, exudative collection, transudate collection, or modified transudate collection), any kind of mass localized in the thorax or mediastinum (abscess, hematoma, or tumor), and traumatic reticuloperitonitis.

The blood was sampled to run a complete biochemical and hematological examination (Appendix A). The results of the biochemical analysis showed a value of plasmatic fibrinogen in the reference range associated with discrete hypoalbuminemia and an increase in total protein concentration, indicative of a non-specific chronic inflammatory syndrome. Hematology showed a white cell count in the upper values of the reference range accompanied by neutrophilia and a switch in the leukocyte ratio (i.e., a concomitant increase in neutrophil granulocytes and a decrease in lymphocyte proportion), which confirmed the previous findings.

An ultrasound examination of the heart and lungs was first performed using a convex probe (MyLab One©, Easote, France Hospimedi, Saint-Crépin-Ibouvillers, France). Using a right-sided thoracic approach, a voluminous echogenic mass was observed (30 cm by 25 cm), severely compressing the right heart, with a reduced size of the right ventricle, accompanied by an anechoic pleural liquid collection (Figure 1). Except for the compression of the right heart described above, the ultrasonography of the heart did not reveal any abnormalities, particularly in the pericardial cavity or the different valves. Ultrasonography of the reticulum was also normal (except for a complete absence of biphasic contraction, in accordance with the clinical signs).

The pleural liquid collection was sampled with a sterile disposable needle (14 gauge, 2 inches) under ultrasound guidance for biochemical and cytological analysis. Macroscopic examination showed a moderately cloudy orange-to-brown liquid. The results were indicative of a non-specific septic exudate (Appendix A).

### 2.2. Final Diagnosis

The thoracic mass was simultaneously sampled under ultrasound guidance and a Tru-Cut needle biopsy (14 gauge, 6 inches, Merit Medical^®^, South Jordan, UT, USA) using a complete aseptic technique accompanied by local anesthesia (procaine chlorhydrate, Procamidor, Axience©, Pantin, France, 15 mL around the area of the puncture).

Small biopsies were routinely processed and examined by light microscopy. One biopsy sampled a moderately cellular mesenchymal neoplasm composed of large polygonal to spindle cells, randomly arranged and supported by a hyalinized collagenous matrix. The neoplastic cells had a pale eosinophilic and finely granular cytoplasm with indistinct cell borders and an oval, variably sized nucleus with a prominent nucleolus (Figure 2). Atypia was marked to severe with anisokaryosis, binucleation, nuclear hyperchromatism, and some giant nuclei. A few mitotic figures were observed among this atypical cell population. The remaining biopsies consisted of either a fibrino-suppurative material, hemorrhage, or dense collagenous connective tissue.

Immunohistochemical staining was performed on paraffin-embedded 3-µm-thick sections (Appendix A). Immunostaining was performed using anti-vimentin V9 (dilution: 1/50; antigenic retrieval: low-pH, M0725, Agilent©, Santa Clara, CA, USA), anti-desmin D33 (1/50, High pH, M0760 Agilent©), anti-actin HHF35 (1/25, M0760, Agilent©), and anti-cytokeratin MNF116 (1/300, Low pH, M0821 Agilent©). Specifically, the immunohistochemical protocol included antigen retrieval for 30 min at 96 °C, except for anti-actin HHFG35, a peroxidase blocking step of 5 min at room temperature (S2023; Agilent©), followed by the saturation of nonspecific binding sites with normal goat serum (X0907; Agilent©) applied for 25 min at room temperature. Primary antibodies were incubated for 50 min at room temperature. Immunohistochemical staining was then visualized using the ENVISIO FLEX HRP system (Agilent©) and 3,3′-diaminobenzidine (DAB) chromogen according to the manufacturer’s recommendations.

All neoplastic cells were diffusely positive for vimentin, variably positive for desmin and actin, and negative for cytokeratin (Figure 3).

Assuming the representativeness of the samples examined, the space-occupying lesions in the thoracic cavity were diagnosed as undifferentiated sarcoma.

Further investigations were performed to explore the potential existence of metastasis elsewhere. Complete biochemical analysis was run and did not reveal any abnormalities in kidney or liver parameters (urea/creatinine and aspartate aminotransferase/creatine kinase/gamma-glutamyl transferase/bilirubin, respectively) (Appendix A). Ultrasound examination (MyLab One©, Easote, France Hospimedi, Saint-Crépin-Ibouvillers, France) was also performed. The peritoneum, liver, intestinal tract, abomasum, and right kidney were evaluated by a transabdominal ultrasound with a convex probe, whereas the reproductive system, urinary bladder, and left kidney were evaluated by a transrectal ultrasound with a linear probe. There were no abnormalities in any of these organs, excluding the detrimental presence of metastasis.

### 2.3. Treatment and Outcome

Complete drainage of the thoracic liquid collection was performed. The skin area, situated in the sixth intercostal space, was aseptically prepared by scrubbing and washing with a chlorhexidine-based soap and a 4% chlorhexidine solution. Under mild sedation (xylazine, Sedaxylan, Dechra©, Montigny-Le-Bretonneux, France, 0.05 mg/kg of body weight, intravenous) and local anesthesia (procaine chlorhydrate, Procamidor, Axience©, Pantin, France, 15 mL around the area of the puncture), the skin halfway between the 4th and 5th thoracic rib was longitudinally incised (surgical steel blade number 22) by 2 cm to facilitate the drainage catheter setting. Then, the drainage catheter (12 mm × 50 cm) was driven into the incision under ultrasound guidance and with a solid-made catheter guide to facilitate the right orientation in the animal until the thoracic liquid collection was ejected through the catheter. Twenty liters of a seropurulent liquid collection was extracted. The catheter was fixed to the skin with a Roman Sandal technique realized with a polydioxanone-based monofilament absorbable suture material (PDS* Plus, dec. 0, Ethicon©, Issy-les-Moulineaux, France). Once the right-sided thoracic cavity was emptied, the latter was flushed twice a day with a 5% chlorhexidine saline solution until the rising liquid was completely clean. This process lasted five days, and the drainage catheter was finally removed.

Concomitantly to the drainage, an antibiotic course based on daily amoxicillin procaine (amoxicillin trihydrate, Vetrimoxin 48 heures, CEVA©, Libourne, France, 15 mg/kg of body weight, intramuscular), was carried out for 12 days. A non-steroidal anti-inflammatory drug (meloxicam, Recocam, Bimeda©, Rennes, France, 0.4 mg/kg of body weight, subcutaneous) was administered on the day of the drainage catheter setting. The day after, the cow received a two-time single corticosteroid injection (Dexalone solution^®^, 0,044 mg/kg) for anti-inflammatory properties and orexigenic effects.

Heart and respiratory rates returned to normal (72 beats per minute and 28 movements per minute, respectively) in the days after the drainage; the appetite was significantly improved, and the cow ate the correct amount of hay (11 kg per day on average) accompanied by 3 kg of concentrates daily, distributed three times (morning, mid-day, and evening).

As the surgical approach for the thoracic sarcoma was difficult, and apparent recovery was shown by the patient, it was discharged after 21 days of hospitalization at the veterinary school. On the basis of the long-term prognosis, the farmer was advised to wait for the end of the antibiotic withdrawal period of 18 days and cull the patient humanely.

A telephone inquiry with the farmer revealed that the cow went to the slaughterhouse after the withdrawal period, with an acceptable body condition score and without any clinical signs between her hospitalization and the culling time. The official veterinarian of the slaughterhouse revealed that the cow had a unilateral right-sided chronic pleurisy with a visible round intra-thoracic 35 cm-diameter tissue-aspect mass between the right heart and the parietal pleura. Unfortunately, no photography was available, and no larger sampling was performed to run a second histopathological analysis for this mass. These post-mortem findings motivated a partial seizure of the right-sided thoracic cavity. After a conscientious examination of the carcass, including both bones and organs, no metastasis was found. Due to the absence of other lesions, the rest of the carcass was deemed fit for human consumption and was put in the food chain.

## 3. Discussion

Bovine oncology has not been given due importance in cattle medicine, even if some conditions, such as viral bovine leukemia, have been studied and regulated for a long time [9]. Being a species with a longer life span, cattle suffer from cancer less frequently than laboratory rodents, for example, but conversely, their lifetime is driven by their economic value and often shortened before the clinical expression of tumors [10,11].

Thoracic tumors in cattle are mainly metastases, but primary tumors exist too [12]. The incidence of such lung tumors among all bovine tumors is 2.8% [6].

Ultrasonography is not commonly used in adult cow respiratory diseases, but this technique has proven efficacious and accurate to diagnose clinical and subclinical pneumonia in dairy calves [13,14,15].

In the present case, pulmonary and cardiac auscultation revealed a marked difference between the right and left sides of the thorax; the noises were less audible on the right side. It was then decided to perform ultrasonography on both sides of the chest. On the right-hand side and in the lower part, the tumor, apparently very close and adherent to the chest wall, pushed back the heart and lung. This location allowed us to see it in its entirety and subsequently try to puncture it with a needle for cytology. Ultrasonography revealed unilateral pleuritis in the upper part of the right lung (picture not shown). As no other abnormalities were detected on clinical examination, and as ultrasonography of the genital tract and the left side of the abdomen did not reveal any other organ damage, we supposed that all troubles were related to the lesioned right lung.

Pulmonary malignant neoplasms are classified according to their origin, either epithelial cells or mesenchymal cells [16,17,18]. “Pseudo-tumors” are represented by hamartomas [19].

In the current case, the identification of proliferative mesenchymal cells in combination with marked cytonuclear atypia, mitotic figures, and necrosis were consistent with malignant neoplasia classified as sarcoma. However, a sarcomatoid form of mesothelioma remained included in the differential histopathological diagnosis at this stage.

Mesotheliomas are tumors of the mesothelial covering cells of the serous membranes and their supporting connective tissue of mesodermal origin. These tumors are well-described in numerous species, including humans and cattle [20,21,22,23,24,25,26]. In veterinary medicine, mesothelioma has been classified into papillary epithelioid, sarcomatoid, and, most commonly, biphasic on the basis of the histological growth pattern [27]. The tumoral cells concern the peritoneum and the pleura and the pericardium exceptionally [28]. In addition, a fluid collection is often reported in the literature, either within the peritoneal cavity and/or within the pleural cavity, leading to respiratory distress [29]. This last element further reinforces the inclusion of mesothelioma in the differential diagnosis, as a fluid collection was demonstrated on the ultrasound of the right-sided thoracic cavity (Figure 1).

Immunochemistry is a useful tool to differentiate the type of tumor. The neoplastic cells were diffusely negative for cytokeratin. Previous cases of mesothelioma in cattle described in the literature showed positivity for cytokeratin [30,31]. As a result, our findings were not in favor of mesothelioma, and the tumor was qualified with confidence as a sarcoma. However, immunochemistry did not allow us to go further in the qualification of the sarcoma with the cell material we collected. The post-mortem collection of the whole tumor for further analysis was impossible, as the owner refused euthanasia.

Sarcomas have a high capacity to generate metastasis in other organs [32,33,34]. Even if the present cow did not show any other clinical signs except for the respiratory disorders, further specific investigations were performed to explore the potential existence of metastasis elsewhere. Tomography is the most sensitive examination to accurately look for the presence of metastasis [35,36]. This technique is available for small species, such as cats and dogs, but it is not adapted for large animals, such as adult cows. Radiography is a current alternative to explore the existence of metastasis [35]. Deep tissue radiography requires powerful generators available in a number of universities, not including the Clinic of Ruminants of Toulouse, France. Investigations were consequently based on ultrasonography and biochemical analysis and regarded the explorable organs in adult ruminants. Complete biochemical analysis was performed and did not reveal abnormalities in other organs (Appendix A). Ultrasonography of all the accessible organs in the abdominal and ultrasound images did not reveal any abnormalities either. As a result, we concluded that no metastasis was present elsewhere, and the lesion was limited to the thoracic cavity. This conclusion was confirmed by a macroscopic examination of the carcass at the slaughterhouse by the onsite veterinarian, who explored all the organs that may be targeted by metastasis, such as the bones and viscera. No lesions were found elsewhere, qualifying the carcass as safe for human consumption.

Specific treatments for sarcomas include surgical excision, radiation therapy, and chemotherapy, as described in other species [37,38]. Surgery, radiation therapy, or their combination are used for local tumor control. Chemotherapy is used to control or prevent metastasis in animals at risk of systemic spread [39]. The surgical procedure requires large and deep margins (2–3 cm) around the sarcoma to make sure that the excision is complete [37,38]. In the present case, the thoracic cavity was hardly accessible in the adult cow, and the achievement of large surgical margins was almost impossible due to the proximity of the heart. For these reasons, the surgical excision was not performed. Radiation therapy and chemotherapy were alternative solutions, but they are costly procedures, and they are hardly applicable in farm animals. Moreover, the clinical state of the cow was improved after treating the complications and the owner preferred to value the carcass at the slaughterhouse as long as the clinical condition was good.

Sarcomas in cattle are very often fatal [8,40,41]. The main reasons are an overall lesion presentation that is too large for successful treatment and/or the massive presence of metastasis elsewhere, making treatment impossible or infructuous [42]. In the present case, the tumor was well-circumscribed within the thoracic cavity. In addition, investigations into the possible existence of metastases did not reveal any lesions elsewhere in the cow. The absence of lesions other than the primary tumor was confirmed at the slaughterhouse, and the carcass could be valued (except for the right-sided thoracic cavity, partially captured, see above), making our case particularly and interestingly original because the outcome was positive for the farmer.

## 4. Conclusions

Primary thoracic neoplasms, although rare, should be considered in the differential diagnosis of respiratory clinical signs and associated weight loss and cachexia in adult cattle. The present case report offers a continuum from the initial clinical signs motivating specific investigations to interesting laboratory findings.

## Figures and Tables

**Figure 1 vetsci-10-00376-f001:**
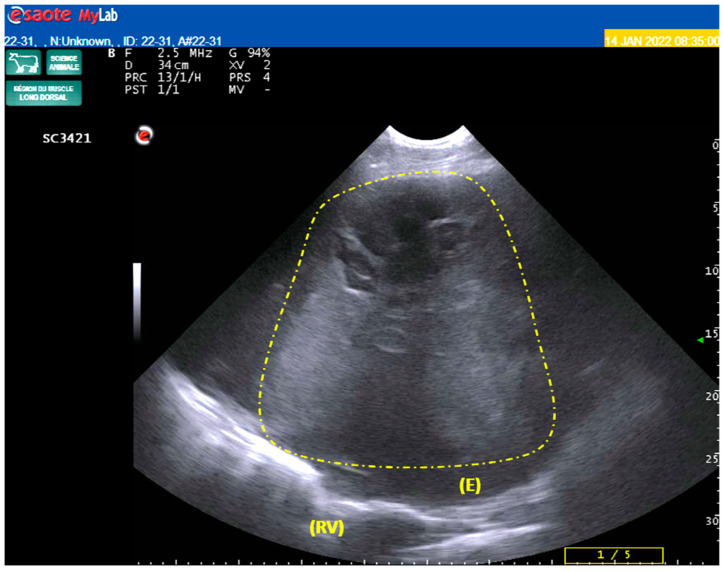
Ultrasonographic aspect of the intra-thoracic right-sided mass (yellow dashed lines), accompanied by anechoic liquid collection (E) and compression of the right ventricle (RV). Source: Clinic of Ruminants, National Veterinary School of Toulouse, Toulouse, France.

**Figure 2 vetsci-10-00376-f002:**
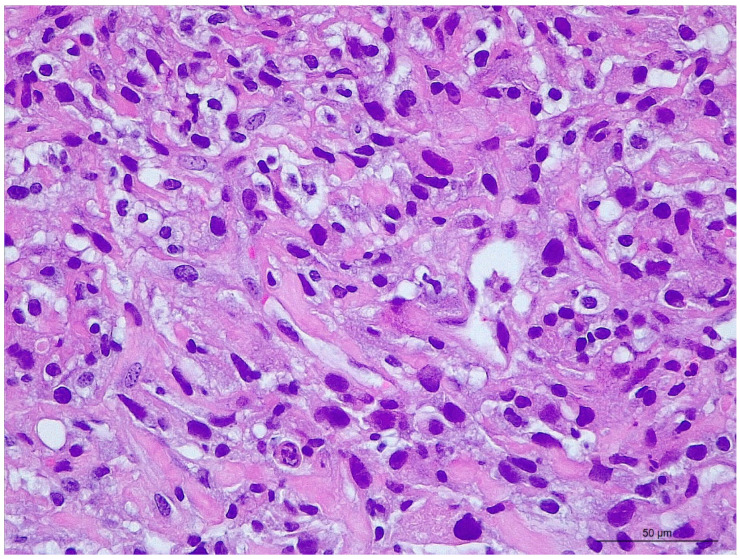
Microscopic examination of the biopsy sampling (hematoxylin and eosin coloration; magnification: ×40). Source: LAPVSO, Toulouse, France; neoplastic tissue consisted of randomly arranged pleomorphic spindle to polygonal mesenchymal cells with marked atypia, supported by a sparse hyalinized collagenous matrix.

**Figure 3 vetsci-10-00376-f003:**
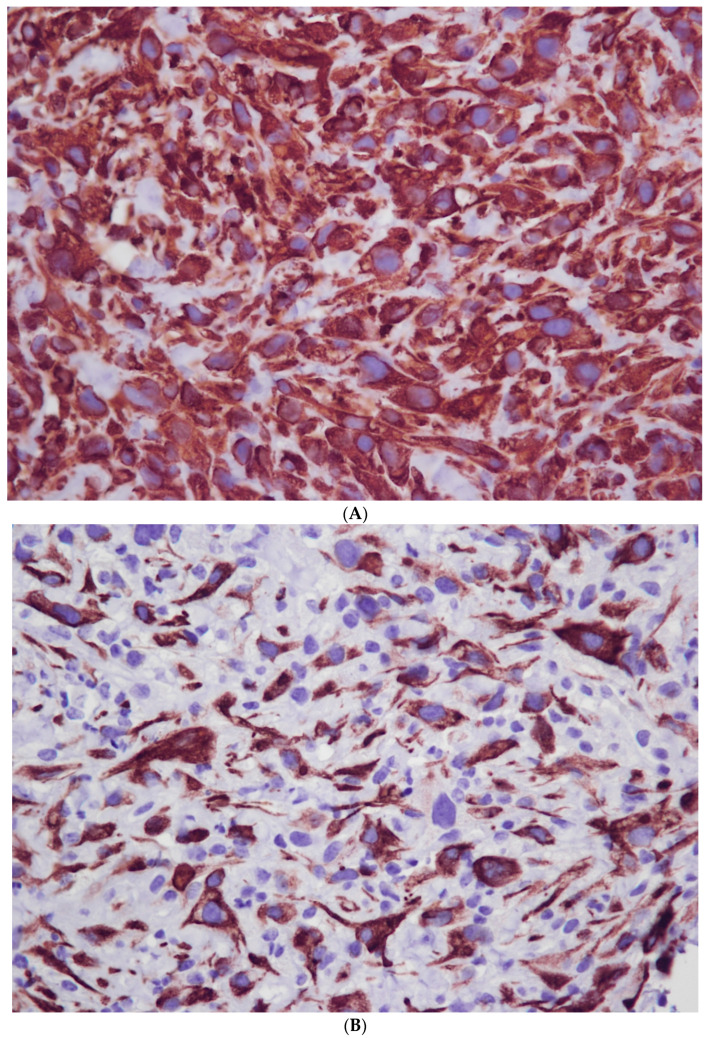
Immunohistochemistry (magnification: 40×). (**A**) Vimentin V9, (**B**) desmin D33; source: Department of Basic Sciences, IHAP, Université de Toulouse, INRAE, ENVT, Toulouse, France; neoplastic cells diffusely expressed vimentin but were variably positive for desmin.

## Data Availability

Not applicable.

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
