# Peer review of "Thoracic Tumor Associated with a Unilateral Empyema in a Beef Cow: A Case Report"

_vetsci, 2023, doi:10.3390/vetsci10060376_

Round 1

Reviewer 1 Report

In lines 268-280, the authors digressed, and discuss BRD. Case reports are presented succinctly. The mentioned lines should be revised or removed. 

In general, the discussion need not be long. 

Author Response

In lines 268-280, the authors digressed, and discuss BRD. Case reports are presented succinctly. The mentioned lines should be revised or removed. 

In general, the discussion need not be long. 

AU: thank you for your pertinent comments. Those lines you mentioned were removed to focus the discussion more specifically on thoracic masses.

Reviewer 2 Report

Reviewer comments for manuscript ID vet sci-240226 entitled ‘Thoracic sarcoma associated with a unilateral empyema in a beef cow: a case report’.

General Comments

Tumorous growths in the thoracic cavity of cattle are rarely diagnosed clinically and are often incidental findings upon necropsy. Diagnosing intrathoracic tumour lesions are a challenge to the clinicians and require advanced diagnostic modalities. The clinical outcome of such lesions is not very positive most of the times, but report of such cases will definitely add to scientific literature and knowledge. This might spur research and strategies to identify the causal factors and preventative strategies. The authors have presented an interesting case of an undifferentiated sarcoma in the thoracic cavity of the cow. The case has been excellently diagnosed through systemic clinical examination, diagnostic imaging and confirmed through a thorough laboratory examination. I congratulate the authors for the hard work.

The manuscript is well written in patches that I feel might be due to the non-native English-speaking background of the authors. The figures and tables add to the richness of the investigation. I need few clarifications from the authors and would like to see changes in the manuscript as per my comments before I recommend the publication of the manuscript.

Specific Comments

Line 10: Please reword this as ‘Tumours in cows have been less frequently reported in literature’.

Lines 10-12: Please reword this as 'They often represent as unusual findings in live animals and incidental at slaughter with a rare positive therapeutic outcome for the farmers’.

Line 12: Please delete ‘In that context’.

Lines 18-20: Please rewrite as ‘After the end of the withdrawal period, the cow recovered clinically but was culled by the owners for economic reasons.

Lines 21-22: Please replace ‘with a final profitable outcome for the owner which is not common for this type of affection’ with ‘that were confirmed upon post-mortem’.

Line 34: Please delete ‘In January 2022’.

Line 38: Please replace ‘taken over’ with ‘treated’.

Lines 40-41: Please replace ‘associated with’ with ‘and’

Line 43: Please replace ‘in Toulouse’ with ‘at the clinic of ruminants of the National Veterinary School of Toulouse, France’.

Lines 48-49: Please replace ‘The cow did not pay attention to proposed concentrates and hay showing a poor appetite’ with ‘The cow revealed complete inappetence’.

Line 51: Please delete ‘for the present cow’.

Line 55: Please replace ‘synonymous of an important’ with ‘indicating a’.

Lines 63-64: Please replace ‘The challenge consisting in applying a skin fold in the withers area turns out to be positive’ with ‘Skin fold pinch test in the wither area was positive’.

Line 65: Please replace ‘meaning’ with ‘indicating’.

Line 66: Please replace ‘manifest’ with ‘manifested’.

Line 67-68: Please reframe as ‘The palpation of the external lymph nodes revealed a moderate hypertrophy of the right superficial cervical lymph node.

Line 69: Please replace ‘about’ with ‘in’.

Line 72: Please replace ‘modification’ with ‘alteration’.

Lines 80-81: How could you arrive at a differential diagnosis of mesothelioma without a histopathology? Please clarify.

Lines 81-82: Please delete this sentence and add ‘traumatic reticulo-peritonitis ‘to the above-mentioned differentials.

Line 86: Please replace ‘which is in favour’ with ‘indicative of’.

Lines 97-102: The ultrasonography finding revealed changes in the right-side heart chambers and then you further say ‘Ultrasonography of the heart did not reveal any abnormalities. Please clarify this ambiguity and rewrite.

Line 110: Please replace ‘in favour’ with ‘indicative’.

Line 115: Please delete ‘thanks to’.

Lines 146-47: Please reframe as ‘Assuming representativeness of the samples examined, the space occupying lesions in the thoracic cavity was diagnosed as an undifferentiated sarcoma.

Lines 157: Please delete ‘In order to get comfort for the cow’.

Lines 160-63: Please reframe as ‘Under mild sedation (xylazine, Sedaxylan©, Dechra, France, 0.05 mg/kg 160 of body weight, intravenous way) and local anaesthesia (procaine chlorhydrate, Procam idor©, Axience, France, 15 mL around the area of puncture), the skin at the level of ­­­­­­­­­­­­­­­­­­­­­­­­­­­­­­­­­­­­­­­­­­_please mention the exact site (rib or intercostal space level and point ), ________________was longitudinally incised (surgical steel blade number 22) by 2 cm to facilitate the drainage catheter setting.

Line 170: Please replace ‘thanks to a’ with ‘with’.

Line 180: Please replace ‘run to normality’ with ‘returned to normal’.

Lines 184-88: Please reframe as ‘As the surgical approach to the thoracic sarcoma was difficult and apparent recovery shown by the patient, it was discharged after 21 days of hospitalization at the veterinary school. Based on the long-term prognosis, the farmer was advised the farmer to wait for the end of antibiotic withdrawal period of18 days and cull the patient humanely.

Line 189: Please reframe as ‘Telephonic news to the’ with ‘Telephonic inquiry with the farmer revealed that the cow went to the slaughterhouse after 3 months from hospitalization’.

Line 201: Please replace ‘is not one of the main parts of’ with ‘has not been given due importance’.

Line 203: Please replace ‘As long-lived animals’ with ‘Being a species with a longer life span’.

Line 209: Please replace ‘efficiency and accuracy’ with ‘efficacious and accurate’.

Line 240: Please replace ‘regarded’ with ‘of’.

Line 242: Please replace ‘issue only regarded’ with ‘the lesion was limited to’

Line 273: Please replace ‘rareness’ with ‘rarity’.

Line 285-86: Please delete ‘with a final profitable outcome for the owner which is not common for this type of affection’.

Respected Editors,

                                I feel the manuscript needs moderate editing of english language that might have arisen due to the non-native english speaking background of the authors.

Thanks

Author Response

General Comments

Tumorous growths in the thoracic cavity of cattle are rarely diagnosed clinically and are often incidental findings upon necropsy. Diagnosing intrathoracic tumour lesions are a challenge to the clinicians and require advanced diagnostic modalities. The clinical outcome of such lesions is not very positive most of the times, but report of such cases will definitely add to scientific literature and knowledge. This might spur research and strategies to identify the causal factors and preventative strategies. The authors have presented an interesting case of an undifferentiated sarcoma in the thoracic cavity of the cow. The case has been excellently diagnosed through systemic clinical examination, diagnostic imaging and confirmed through a thorough laboratory examination. I congratulate the authors for the hard work.

The manuscript is well written in patches that I feel might be due to the non-native English-speaking background of the authors. The figures and tables add to the richness of the investigation. I need few clarifications from the authors and would like to see changes in the manuscript as per my comments before I recommend the publication of the manuscript.

Specific Comments

Line 10: Please reword this as ‘Tumours in cows have been less frequently reported in literature’.

AU: Done.

Lines 10-12: Please reword this as 'They often represent as unusual findings in live animals and incidental at slaughter with a rare positive therapeutic outcome for the farmers’.

AU: Done.

Line 12: Please delete ‘In that context’.

AU: Done.

Lines 18-20: Please rewrite as ‘After the end of the withdrawal period, the cow recovered clinically but was culled by the owners for economic reasons.

AU: Done.

Lines 21-22: Please replace ‘with a final profitable outcome for the owner which is not common for this type of affection’ with ‘that were confirmed upon post-mortem’.

AU: Done.

Line 34: Please delete ‘In January 2022’.

AU: Done.

Line 38: Please replace ‘taken over’ with ‘treated’.

AU: Done.

Lines 40-41: Please replace ‘associated with’ with ‘and’

AU: Done.

Line 43: Please replace ‘in Toulouse’ with ‘at the clinic of ruminants of the National Veterinary School of Toulouse, France’.

AU: Done.

Lines 48-49: Please replace ‘The cow did not pay attention to proposed concentrates and hay showing a poor appetite’ with ‘The cow revealed complete inappetence’.

AU: Done.

Line 51: Please delete ‘for the present cow’.

AU: Done.

Line 55: Please replace ‘synonymous of an important’ with ‘indicating a’.

AU: Done.

Lines 63-64: Please replace ‘The challenge consisting in applying a skin fold in the withers area turns out to be positive’ with ‘Skin fold pinch test in the wither area was positive’.

AU: Done.

Line 65: Please replace ‘meaning’ with ‘indicating’.

AU: Done.

Line 66: Please replace ‘manifest’ with ‘manifested’.

AU: Done.

Line 67-68: Please reframe as ‘The palpation of the external lymph nodes revealed a moderate hypertrophy of the right superficial cervical lymph node.

AU: Done.

Line 69: Please replace ‘about’ with ‘in’.

AU: Done.

Line 72: Please replace ‘modification’ with ‘alteration’.

AU: Done.

Lines 80-81: How could you arrive at a differential diagnosis of mesothelioma without a histopathology? Please clarify.

AU: your comment is very welcome. We discuss about the possibility of mesothelioma further in the manuscript at a more appropriate place, please see lines 256 – 272.

Lines 81-82: Please delete this sentence and add ‘traumatic reticulo-peritonitis ‘to the above-mentioned differentials.

AU: Done.

Line 86: Please replace ‘which is in favour’ with ‘indicative of’.

AU: Done.

Lines 97-102: The ultrasonography finding revealed changes in the right-side heart chambers and then you further say ‘Ultrasonography of the heart did not reveal any abnormalities. Please clarify this ambiguity and rewrite.

AU: thank you for your comment. Some details were added in the manuscript (please see lines 117 – 199 and below)

Except the compression of the right heart described above, the ultrasonography of the heart did not reveal any abnormalities, particularly in the pericardial cavity or the different valves.”

Line 110: Please replace ‘in favour’ with ‘indicative’.

AU: Done.

Line 115: Please delete ‘thanks to’.

AU: Done.

Lines 146-47: Please reframe as ‘Assuming representativeness of the samples examined, the space occupying lesions in the thoracic cavity was diagnosed as an undifferentiated sarcoma.

AU: Done.

Lines 157: Please delete ‘In order to get comfort for the cow’.

AU: Done.

Lines 160-63: Please reframe as ‘Under mild sedation (xylazine, Sedaxylan©, Dechra, France, 0.05 mg/kg 160 of body weight, intravenous way) and local anaesthesia (procaine chlorhydrate, Procam idor©, Axience, France, 15 mL around the area of puncture), the skin at the level of ­­­­­­­­­­­­­­­­­­­­­­­­­­­­­­­­­­­­­­­­­­_please mention the exact site (rib or intercostal space level and point ), ________________was longitudinally incised (surgical steel blade number 22) by 2 cm to facilitate the drainage catheter setting.

AU: Done.

Line 170: Please replace ‘thanks to a’ with ‘with’.

AU: Done.

Line 180: Please replace ‘run to normality’ with ‘returned to normal’.

AU: Done.

Lines 184-88: Please reframe as ‘As the surgical approach to the thoracic sarcoma was difficult and apparent recovery shown by the patient, it was discharged after 21 days of hospitalization at the veterinary school. Based on the long-term prognosis, the farmer was advised the farmer to wait for the end of antibiotic withdrawal period of18 days and cull the patient humanely.

AU: Done.

Line 189: Please reframe as ‘Telephonic news to the’ with ‘Telephonic inquiry with the farmer revealed that the cow went to the slaughterhouse after 3 months from hospitalization’.

AU: Done.

Line 201: Please replace ‘is not one of the main parts of’ with ‘has not been given due importance’.

AU: Done.

Line 203: Please replace ‘As long-lived animals’ with ‘Being a species with a longer life span’.

AU: Done.

Line 209: Please replace ‘efficiency and accuracy’ with ‘efficacious and accurate’.

AU: Done.

Line 240: Please replace ‘regarded’ with ‘of’.

AU: Done.

Line 242: Please replace ‘issue only regarded’ with ‘the lesion was limited to’

AU: Done.

Line 273: Please replace ‘rareness’ with ‘rarity’.

AU: Done.

Line 285-86: Please delete ‘with a final profitable outcome for the owner which is not common for this type of affection’.

AU: Done.

 I feel the manuscript needs moderate editing of english language that might have arisen due to the non-native english speaking background of the authors.

AU: Thank you for your advises. We have revised the manuscript to address the issues raised by the reviewer, including moderate editing of the English language. We hope that these changes have addressed your concerns and improved the quality of our work. 

Reviewer 3 Report

The paper entitled: thoracic sarcoma associated with a unilateral empyema in a beef cow: a case report, describes a clinicopathological investigation of a thoracic mass discovered in a 9-year-old cow. Ultrasonography of the thorax and immunohistochemistry in small biopsies allow for the diagnosis of a thoracic sarcoma severely compressing the right heart. Tumors in the thorax are not very frequent in cows and how to differentiate these tumors from other lesions is relevant for veterinary practitioners.

Several comments/suggestions are listed in the following:

-Introduction is too short and more details about thoracic masses or pathologies should be provided. Thoracic tumors differentiate from other lesions like inflammatory origin.

-Clinical details (lines 44-69)need to be summarized. Many details that are not very relevant may disturb the reader's concentration.

-Lines 76-81 should go to the discussion section.

-Table 1 contains data not relevant and the important information is detailed on lines 83-90. Table 1 should be deleted.

-Pathology results should be described in a separate section and then describe treatment and outcome.

-Line 132. The immunohistochemical procedure needs to be described shortly or include a reference. I suggest including other tumor markers of the connective tissue tumors. A panel of antibodies differentiating mesotheliomas used in the paper is sufficient in the opinion of this reviewer.

-Figure 3 would need more clarity they are too dark.

-Lines 156-188 are too long and shortened version is recommended.

-Postmortem information provided by telephone may not be the best way to confirm data.

-Discussion section is too speculative and should be concentrated on the thoracic masses, particularly tumors. In my opinion, authors should be concentrated on explaining with clarity why this thoracic mass is not inflammatory and why it is a sarcoma, not other thoracic tumors or pathologies. Mesotheliomas should be a major discussion point.

-I think there are many poorly used references or not used.

Author Response

The paper entitled: thoracic sarcoma associated with a unilateral empyema in a beef cow: a case report, describes a clinicopathological investigation of a thoracic mass discovered in a 9-year-old cow. Ultrasonography of the thorax and immunohistochemistry in small biopsies allow for the diagnosis of a thoracic sarcoma severely compressing the right heart. Tumors in the thorax are not very frequent in cows and how to differentiate these tumors from other lesions is relevant for veterinary practitioners.

Several comments/suggestions are listed in the following:

-Introduction is too short and more details about thoracic masses or pathologies should be provided. Thoracic tumors differentiate from other lesions like inflammatory origin.

AU: thank you for your interesting comment. Inflammatory and neoplastic conditions of the thoracic cavity in cattle were detailed in the introduction, lines XXX. This brief of overview was supported the references added in the revised manuscript.

-Clinical details (lines 44-69) need to be summarized. Many details that are not very relevant may disturb the reader's concentration.

AU: some details were removed. All remaining clinical data are consistent with the clinical approach for animal examination and they highlight major abnormalities such as pain with predominantly thoracic symptoms, with a qualitative estimate of the time course.

-Lines 76-81 should go to the discussion section.

AU: The differential diagnosis is placed here because it links the important data collected during the clinical examination and the subsequent ancillary examinations which were performed in order to confirm or invalidate the hypotheses.

-Table 1 contains data not relevant and the important information is detailed on lines 83-90. Table 1 should be deleted.

AU: thank you for your comment. Figures from blood tests were removed and will be available as supplemental material. 

-Pathology results should be described in a separate section and then describe treatment and outcome.

AU: Thank you for this comment. Appropriate separated sections were put.

-Line 132. The immunohistochemical procedure needs to be described shortly or include a reference. I suggest including other tumor markers of the connective tissue tumors. A panel of antibodies differentiating mesotheliomas used in the paper is sufficient in the opinion of this reviewer.

AU: Thank you for your comment. Immunohistochemical procedure was briefly detailed lines XXX, including clones, dilutions, incubation and antigenic retrieval conditions applied for the detection of Vimentin, Cytokeratin, Desmin, Actin. Unfortunately, our diagnostic service does not have additional available and validated markers for the bovine species. We understand that the addition of other tumor markers would have enriched the case. However, the provided antigenic immunophenotype remains helpful for the exclusion of serosal neoplastic conditions reported in cattle such as mesothelioma, carcinomatosis.

-Figure 3 would need more clarity they are too dark.

AU: Done.

-Lines 156-188 are too long and shortened version is recommended.

AU: according to the authors, providing details therapeutic protocol is important so that such a manuscript can be useful to any veterinarian facing a potential similar case in his own practice.

-Postmortem information provided by telephone may not be the best way to confirm data.

AU: thank you for your comment. The authors are fully aware of the limits of consideration of such information. However, for economic reasons, the cow was sent to slaughter without any further investigations of the carcass. The information provided by the veterinarian in charge of the post-mortem regulatory health inspection confirmed the presence of a thoracic neoplastic mass and the absence of any grossly visible distant metastases.

-Discussion section is too speculative and should be concentrated on the thoracic masses, particularly tumors. In my opinion, authors should be concentrated on explaining with clarity why this thoracic mass is not inflammatory and why it is a sarcoma, not other thoracic tumors or pathologies. Mesotheliomas should be a major discussion point.

AU: thank you for your interesting remark. Some sentences taking into account your suggestion were added in the manuscript, please see lines 254 – 275.

-I think there are many poorly used references or not used.

AU: Following your helpful comment, the authors made a significant effort to correct the use of available references about the topic.

Round 2

Reviewer 3 Report

The paper entitled : thoracic sarcoma associated with unilateral empyema in a cow : a case report , describes a clinicaptological investigation of a mass compressing of the rigth heart of 9-year-old beef cow. the paper describes in detail clinical data and ultrasonography investigation together with small biopsy samples assessment. The paper is well-written and contains useful potential information for veterinary practitioners.

The main inconvenience, under the opinion of this reviewer, is the mass denomination as a sarcoma. Not clear evidence is demonstrated supporting the classification of the tumor as a sarcoma, probably due to the size of the biopsies and the concurrent presentation of purulent inflammation. The pieces of evidence in the pictures and descriptions do not support enough this denomination. As a necropsy study could not be done, I suggest a change to the title accordingly.

Immunohoistochemical pictures are low quality and should be improved.

Author Response

The paper entitled: thoracic sarcoma associated with unilateral empyema in a cow: a case report , describes a clinicopathological investigation of a mass compressing of the right heart of 9-year-old beef cow. the paper describes in detail clinical data and ultrasonography investigation together with small biopsy samples assessment. The paper is well-written and contains useful potential information for veterinary practitioners.

The main inconvenience, under the opinion of this reviewer, is the mass denomination as a sarcoma. Not clear evidence is demonstrated supporting the classification of the tumor as a sarcoma, probably due to the size of the biopsies and the concurrent presentation of purulent inflammation. The pieces of evidence in the pictures and descriptions do not support enough this denomination. As a necropsy study could not be done, I suggest a change to the title accordingly.

AU: According to your suggestion, the authors have accordingly modified the title of the manuscript

Immunohistochemical pictures are low quality and should be improved.

AU: The authors captured higher-quality photographs and included them in the manuscript